# Improvement of Distance Measurement Based on Dispersive Interferometry Using Femtosecond Optical Frequency Comb

**DOI:** 10.3390/s22145403

**Published:** 2022-07-20

**Authors:** Qiong Niu, Mingyu Song, Jihui Zheng, Linhua Jia, Junchen Liu, Lingman Ni, Ju Nian, Xingrui Cheng, Fumin Zhang, Xinghua Qu

**Affiliations:** State Key Laboratory of Precision Measurement Technology and Instruments, Tianjin University, Tianjin 300072, China; qongniu@163.com (Q.N.); mingyu.song@hirain.com (M.S.); jihui.zheng@foxmail.com (J.Z.); jialinhua@tju.edu.cn (L.J.); liujunchen_jyxy@tju.edu.cn (J.L.); nilingman@tju.edu.cn (L.N.); nianju@tju.edu.cn (J.N.); cheng.x.r@foxmail.com (X.C.); quxinghua@tju.edu.cn (X.Q.)

**Keywords:** absolute distance measurement, optical frequency comb, dispersive interferometry, Lomb–Scargle algorithm

## Abstract

Since the dispersive interferometry (DPI) based on optical frequency combs (OFCs) was proposed, it has been widely used in absolute distance measurements with long-distance and high precision. However, it has a serious problem for the traditional DPI based on the mode-locked OFC. The error of measurements caused by using the fast Fourier transform (FFT) algorithm to process signals cannot be overcome, which is due to the non-uniform sampling intervals in the frequency domain of spectrometers. Therefore, in this paper, we propose a new mathematical model with a simple form of OFC to simulate and analyze various properties of the OFC and the principle of DPI. Moreover, we carry out an experimental verification, in which we adopt the Lomb–Scargle algorithm to improve the accuracy of measurements of DPI. The results show that the Lomb–Scargle algorithm can effectively reduce the error caused by the resolution, and the error of absolute distance measurement is less than 12 μm in the distance of 70 m based on the mode-locked OFC.

## 1. Introduction

With the development of science and technology, distance measurement with higher precision is widely used in modern industry and scientific research, such as industrial production, measurement of gravitational waves, acquisition of black-hole images, and observation of satellite formations. However, due to the current technical level, it is very challenging to realize distance measurement with a large length and high accuracy at the same time, so it is urgent to find a method with high-precision and long-distance range. 

The emergence of OFCs has opened up a new way to solve this problem. The related concepts were first proposed by Hänsch in 1978 [1]. OFCs are a kind of pulse-light that have a time width in femtoseconds [2]. They provide a new research direction for so many fields of distance measurement, precision spectroscopy frontier physics, and so on [3,4]. They have been closely related to the development of the country and people’s lives.

In 2000, Minoshima and Matsumoto of the National Institute of Metrology in Japan applied an OFC for absolute distance measurement for the first time [5]. This kind of distance measurement method based on OFCs includes synthetic wavelength interferometry [6,7,8], multi-wavelength interferometry [9], time-of-flight [10], dispersive interferometry [11], and optical sampling method [12].

In 2006, Joo and Kim et al. proposed a scheme for absolute distance measurement by DPI, and achieved a 1.46 mm non-ambiguity range and a resolution of 7 nm in 0.89 m [13,14], which also ignored the error brought by the non-uniform frequency sampling of the spectrometer. In 2013, Shujian Xing used a combination of multi-wavelength interferometry and time-of-flight based on a Michelson interferometer to conduct distance measurement within a range of 60 cm with a measurement error of 0.5 μm [15], though it had to guarantee the frequency stability of multiple wavelengths to form a synthetic wavelength chain, leading to a decrease of measurement speed. In 2016, Hanzhong Wu et al. used cavity tuning for optical sampling to achieve a large distance measurement with an error of 10 μm within 50 m [16], but frequency sweeping by cavity tuning limited the characteristics of real-time measurement and led to a complex system architecture and high maintenance costs.

In addition, the method of measurement based on mode-locked OFCs has high accuracy and excellent stability, and the traditional FFT algorithm is often used for data processing in DPI [17]. However, it will produce a larger error when the signal–noise ratio (SNR) is low, and reference [17] has confirmed the fragility of the classical FT method in the measurements of an interference signal with high phase noise. Moreover, due to the wide spectrum of OFCs, the frequency sampling interval of the spectrometer will be unequal, and new errors will be introduced. Therefore, we analyze the mathematical model and properties of the optical frequency comb, and two distance measurements are carried out by using DPI with a mode-locked OFC. Moreover, we also adopt a new method, the Lomb–Scalgle algorithm, to process the data, which is suitable for the analysis of various non-uniform sampling data and has been widely used in the astronomy community.

## 2. Principles and Methods

### 2.1. A Simplified Mathematical Model for Analyzing the Characteristics of the OFC

To get a mathematical expression of the OFC theoretically, we assume that the intensity of each frequency component is 1, the initial phase of each frequency component is 0, and the pulse propagates in a vacuum. In an electric field, the expression of the OFC with *M* harmonics can be written as follows:(1)$$E(t)=\sum_{m=0}^{M}\cos[(f_{0}+mf_{r})2\pi t].$$

Here, *f_r_* is the repetition rate of the OFC, *f*_0_ denotes the carrier-envelope-offset (CEO) frequency, and *m* is an integer. The center frequency of the comb can be calculated as *f_c_* = *f*_0_ + *Mf_r_*/2. Here, the derivation of *E*(*t*) is divided into two different cases: *M* is odd and *M* is even.

#### 2.1.1. M Is Even

When *M* is even, the center frequency *f_c_* will overlap with the *M*/2-order harmonic of the OFC, and Equation (1) can be rewritten as
(2)$$E(t)=\sum_{m=-\frac{M}{2}}^{\frac{M}{2}}\cos[(f_{c}+mf_{r})2\pi t],$$

When *m*_1_ and *m*_2_ are different values of M, and *m*_1_ + *m*_2_ = 0, we can obtain
(3)$$\cos[2\pi t(f_{c}+m_{1}f_{r})]+\cos[2\pi t(f_{c}+m_{2}f_{r})]=2\cos(2\pi f_{c}t)\cos(2\pi mf_{r}t).$$

By using the above equation, Equation (2) can be simplified as
(4)$$E(t)=\cos(2\pi f_{c}t)+\sum_{m=1}^{\frac{M}{2}}2\cos(2\pi f_{c}t)\cos(2\pi mf_{r}t)=\cos(2\pi f_{c}t)\times (1+2\sum_{m=1}^{\frac{M}{2}}\cos(2\pi mf_{r}t)),$$
and the sum of the cosine sequence can be calculated as follows:(5)$$\sum_{m=1}^{\frac{M}{2}}\cos(2\pi mf_{r}t)=\frac{\sin[(M+1)\pi f_{r}t]-\sin(\pi f_{r}t)}{2\sin(\pi f_{r}t)}=\frac{\sin[(M+1)\pi f_{r}t]}{2\sin(\pi f_{r}t)}-\frac{1}{2}=\frac{E_{m}(t)-1}{2},$$
(6)$$E_{m}(t)=\frac{\sin[(M+1)\pi f_{r}t]}{\sin(\pi f_{r}t)},$$
where *E_m_*(*t*) is defined as the modulated signal of the OFC. By combining Equations (4) and (5), the expression of the OFC can be derived as
(7)$$E(t)=\cos(2\pi f_{c}t)\times E_{m}(t).$$

#### 2.1.2. M Is Odd

When *M* is odd, the center frequency does not equal to any frequency component of OFC. To connect this situation with the situation where *M* is even, we express the *E*(*t*) as follows:(8)$$E(t)=E_{1}(t)-E_{2}(t),$$

As shown above, we define two auxiliary signals *E*_1_(*t*) and *E*_2_(*t*): *E*_1_(*t*) is an OFC with a repetition frequency of *f_r_*/2, and it has 2*M* + 1 different frequency components, while *E_2_*(*t*) is an OFC with a repetition frequency of *f_r_*, and it has *M* different frequencies. The signals *E*_1_(*t*) and *E*_2_(*t*) are similar to *E*(*t*) shown in Equation (2); as a result, *E*_1_(*t*) and *E*_2_(*t*) can be simplified as follows:(9)$$E_{1}(t)=\sum_{m=0}^{2M}\cos[(f_{0}+\frac{mf_{r}}{2})2\pi t]=\cos(2\pi f_{c}t)\times \frac{\sin[(2M+1)\pi \frac{f_{r}}{2}t]}{\sin(\pi \frac{f_{r}}{2}t)},$$
(10)$$E_{2}(t)=\sum_{m=0}^{M-1}\cos[(f_{0}+\frac{f_{r}}{2}+mf_{r})2\pi t]=\cos(2\pi f_{c}t)\times \frac{\sin(M\pi f_{r}t)}{\sin(\pi f_{r}t)}.$$

By using the above two equations, we can obtain *E*(*t*) from Equation (8) as follows:(11)$$E(t)=\cos(2\pi f_{c}t)\times \frac{\sin[(2M+1)\frac{p}{2}]\sin(p)-\sin(Mp)\sin(\frac{p}{2})}{\sin(p)\sin(\frac{p}{2})},$$
where $p=\pi f_{r}t$. The modulated signal can be simplified as follows:(12)$$E_{m}(t)=\frac{\sin[(M+1)p]}{\sin(p)}.$$

By using Equation (12), the expression of the OFC *E*(*t*) in Equation (8) can be rewritten as Equation (7), which indicates that the value of *M* cannot influence the form of *E*(*t*), and Equation (7) can be used as the mathematical model of the OFC.

### 2.2. Characteristics of the Frequency Comb

By analyzing the expression (11) and (12), we can know that *E*(*t*) is composed of a modulated signal and a carrier signal. Moreover, the modulated signal *E_m_*(*t*) determines the characteristics of *E*(*t*). The intensity of *E*_1_(*t*) is equal to the intensity of *E_m_*(*t*), and we can obtain its peak intensity by calculating the limit of its modulated signal when *p = πk* (*k* is an integer number).
(13)$$I={\left(\underset{p\to k\pi }{\lim}\frac{\sin[(M+1)p]}{\sin(p)}\right)}^{2}={\left(M+1\right)}^{2}.$$

Ideally, the peak intensity of the OFC depends on the number of its frequency components only. The modulated signal is periodic, and we can get its period by calculating the distance of two neighbouring crests. From Equation (13), the period of the modulated signal is Δ*p* = *πf_r_*·Δ*t* = *π*, and the period *T* can be derived as *T =* 1/*f_r_*. The width of the pulse in the time domain can be derived from Equation (6). When sin[(*M +* 1)*πf_r_t*] *=* 0 and sin(*πf_r_t*) ≠ 0, there are two closest zero-crossing points *z*_1_ and *z*_2_ on both sides of the crest. The distance of these two points can be calculated as follows:(14)$$W=\frac{2}{(M+1)f_{r}},$$
where *W* can be used to estimate the width of the OFC in the time domain.

As shown in Figure 1, we can obtain the simulated signal based on Equation (7). Here, the value of *M* is 381, *f_r_* = 10 GHz, and *f_c_* = 194.2 THz. The amplitude of the pulse peak is 382 and the period of the pulse train is 0.1 × 10^−9^ s, which meets the conclusions of Equation (13). Moreover, the single pulse circled by the red box is also shown in Figure 1b and, according to Equation (14), the full width at half maximum of the pulse in the time domain is about 5.24 × 10^−13^ s.

### 2.3. The Dispersive Interferometry

#### 2.3.1. The Principle of Dispersive Interferometry

The intensity of the interference signal can be written as
(15)$$I(f)=|E_{rp}(f)+E_{mp}(f){|}^{2},$$
where *E_rp_*(*t*) and *E_mp_*(*t*) denote the Fourier transform of the reference pulse train (*E_rp_*(*t*)) and the measurement pulse train (*E_mp_*(*t*)), respectively. Here, we assume that the measurement signal and reference signal have the same amplitude, the measurement signal can be considered as the reference signal with a time delay *τ*, and it can be written as *E_mp_*(*t*) = *E_rp_*(*t* − *τ*). According to the time shift property of Fourier transform, the relation of these two signals can be shown as follows:(16)$$E_{mp}(f)=E_{rp}(f)\times e^{-j2\pi f\tau }.$$

Furthermore, the interference signal *I*(*f*) can be simplified as
(17)$$I(f)=2E_{rp}{\left(f\right)}^{2}+2E_{rp}{\left(f\right)}^{2}\cos(2\pi f\tau ).$$

In the frequency domain, *I*(*f*) is modulated by a signal whose frequency is *τ*. The distance difference *L* between the reference signal and the measurement signal can be calculated as *L* = *τc*/2.

Based on the conclusion of Equation (7), we obtain two simulated signals in the time domain, *E_rp_*(*t*) and *E_mp_*(*t*). To avoid the distortion of the modulated signal which is extracted by selecting the peak value of each frequency component of the interference signal, we set *E_mp_*(*t*) = 2*E_rp_*(*t − τ*), so that the modulation depth of the interference signal can be shallower and the peak values are above zero. The interference signal *I*(*f*) is shown in Figure 2. The values of the center frequency *f_c_*, repetition frequency *f_r_*, and *M* used in our program are 194.2 THz, 10 GHz, and 381, respectively. The time difference between the reference comb and the measurement comb is 4 × 10^−12^ s, as shown in Figure 3b, and the frequency of the modulated signal is 4 × 10^−12^ Hz, which is equal to the value of the time delay.

#### 2.3.2. The Resolution of Dispersive Interferometry

The absolute distance measurement with dispersive interferometry depends on the frequency resolution of the interference signal. According to the Nyquist condition for sampling, the resolution of the time delay can be shown as follows:(18)$$\Delta \tau =\frac{F_{s}}{N}=\frac{1}{N\Delta t}=\frac{1}{B},$$
where *B* is the spectral width of the frequency comb, *F_s_* is the sampling frequency, and Δ*t* is the sampling interval. Based on equality *L* = *τc*/2, the resolution of distance can be shown as follows:(19)ΔL=c/2B.

Thus, we can conclude that the spectral width of the frequency comb decides the resolution of the dispersive interferometry: the wider spectrum of the frequency comb, the higher the resolution. 

The simulation of measurement resolution is shown in Figure 4. The values of the center frequency *f_c_*, repetition frequency *f_r_*, and *M* used in our program are 194.2 THz, 10 GHz, and 381, respectively, and thus the spectral width of the OFC is 3.8 THz. According to Equation (19), we can obtain the resolution of distance as 0.078 mm, which conforms to the simulation results.

#### 2.3.3. Non-Ambiguity Range of Dispersive Interferometry

The sampling interval of the interference signal and each frequency width of the optical frequency comb decides the measurement range of distance. The spectral width of the frequency comb and the resolution of the spectrometer also decide the sampling interval. When the spectrometer can distinguish each frequency of the frequency comb, the frequency comb can be seen as a sampled signal with an equal sampling interval in the frequency domain. For restoring the sampled signal correctly, the sampling frequency must be two times higher than the highest frequency in the sampled signal. The frequency comb can be seen as a signal with the sampling frequency 1/*f_r_*, so the highest frequency of the restored signal is 1/2*f_r_*. The non-ambiguity range is shown as follows:(20)$$L_{NAR}=c/2f_{r}=L_{pp}/2,$$
where *L_NAR_* is the non-ambiguity range and *L_pp_* is the optical path difference between two adjacent pulses of the OFC. Because the width of each frequency mode of OFC affects the coherence of the measurement light and the reference light, the coherence length *L_c_* determined by the linewidth of each frequency can be expressed by the following equation:(21)$$L_{c}=c/2\Delta f.$$

The Δ*f* is the linewidth of each frequency. To prevent a measurement dead-zone we must meet *L_c_* > *L_NAR_*. Thus, we can adjust the size of *f_r_* to realize the absolute distance measurement without a dead zone. The simulation of the non-ambiguity range is shown in Figure 5:

According to Equation (20), the time delay is 15 mm, which is equal to the result of the simulation. We can assume the measurement result is *L*. When no frequency mixing occurs, such as in point *A*, the real value *L*_1_ is equal to *L*. However, in point *B* the measurement result is *L*_2_ and frequency mixing occurs, so the real value is in point *D*. In point *C* the measurement result is *L*_3_ and frequency mixing also occurs, so the real value is in point *E*. We can use the following equations to obtain the real value:(22)$$L_{2}=N\frac{L_{pp}}{2}+(\frac{L_{pp}}{2}-L_{x}),$$
(23)$$L_{3}=N\frac{L_{pp}}{2}+L_{x},$$
(24)$$N=round(\frac{2l}{L_{pp}}),$$
where *l* is the optical path difference between the measuring signal and the reference signal, *N* is an integer which is obtained by rounding down the quotient of the distance to be measured and *L_pp_*/2, and *L_x_* is the calculation results.

In actual calculation, the derivative of the curve near point *B* is less than 0. When the actual distance increases, the measurement results decrease. The derivative of the curve at point *C* is greater than 0. When the actual distance increases, the measurement results increase. Therefore, Equation (22) or Equation (23) can be selected according to the measurement results at a certain point and the measurement results nearby.

Since the light travels back and forth in the optical path, the actual measurement length is magnified to 2 times. For the convenience of expression, the resolution and non- ambiguity range described below refer to the measurement length of 1 time.

## 3. Experiment and Results

### 3.1. Experiment with the Mode-Locked Frequency Comb

Figure 6 shows the layout of the dispersive interferometry experiment. The light source was a One-Five Origami-15 and the values of the center frequency *f_c_* and repetition frequency *f_r_* used in our program were 192.3 THz and 250 MHz, respectively, while the spectral width of the frequency comb was 15 THz. The spectrometer was a YOKOGAWA AQ6370D. The distance resolution of the system was about 10 μm, and the non-ambiguity range was about 0.6 m.

The long distance absolute measurement experiment was carried out on an 80 m underground optical guide rail of the National Institute of Metrology, China. The tested mirror was placed on the air floatation vehicle and moved steadily on the guide rail. In this experiment, a number of different measuring points were selected for measurement across a long distance, and the measurement result of the interferometer was used as the standard value. At the same time, the field environmental parameters were measured. In the measurement process, the ambient temperature was 21.20 °C, the pressure was 1016.10 hPa, the air humidity was 25.17%, and the air refractive index *n_g_* obtained by the Edlen equation was 1.00026799.

In the experiment, the laser interferometer (Agilent 5519b) was set to zero at the initial position, and the frequency-domain interference signal of the frequency comb was recorded. The corresponding distance was recorded as *L*_0_. During the experiment, the measured mirror was moved continuously, several measuring points were selected for measurement, and the interferometer’s signal and the interference signal of the frequency comb were recorded at the same time. If the result of the interference signal of the frequency comb is *L_D_*, the optical path difference between the measured position and the starting position can be expressed as
(25)$$L_{R}=\frac{L_{D}-L_{0}}{2}.$$

### 3.2. Lomb–Scargle Algorithm and Experimental Results

The interference spectra at different positions were obtained in the measurement process as shown in Figure 7. The data in the upper left corner are the distance between the measurement signal and the reference signal, and the data in the upper right corner are the measurement result obtained by the interferometer. The larger the interval between the two signals, the higher the carrier frequency of the interference signal and the denser the interference fringes.

Because the spectrometer can only collect the interference signal by equal wavelength interval sampling in the measurement process, if the traditional time-frequency analysis method is used in the demodulation process of the interference signal carrier frequency, it will bring theoretical error.

Next, the error is analyzed simply according to the equation *c* = *λf*, where *c* is the speed of light, *λ* is the wavelength, and *f* is the optical frequency. Since the sampling interval of the spectrometer is equal to the sampling interval for the wavelength, and assuming the sampling interval of its wavelength is Δ*λ*, the corresponding frequency sampling interval can be expressed by the following equation:(26)$$\Delta f=\frac{f\Delta \lambda }{\Delta \lambda +\lambda }.$$

According to the Equation (26), when the spectrometer meets the equal wavelength interval sampling, the frequency interval sampling is not even. Although the traditional Fourier transform analysis method works well in processing continuous and uniform sampling time series, so that we could obtain a perfect spectrum, when dealing with the actual observation data that are non-uniformly sampled and contain a lot of noise, due to the non-uniformity of the data interval and finite time span, the power spectrum of Fourier transform would contain noise too, which would lead to the reduction of the characteristic spectrum power and the generation of a false spectral peak value of the signal. In addition, due to the enhancement of the noise signal, the amplitude and phase of the real periodic signal will be greatly affected, resulting in certain deviation, and in serious cases it will lead to failure of the parameter extraction. At the same time, to overcome the limitations in resolution, we adapted the Lomb–Scargle algorithm, which is based on the discrete Fourier transform principle: the time sequence is decomposed into a linear combination of some sinusoidal functions Y = A cos(ωt) + B sin(ωt), and then, the characteristics of the signal in the time domain can be transformed into the frequency domain.

In the Lomb–Scargle algorithm, the model curve of the data is fitted by using the sine function and least square method and the root mean square error (RMSE) is used to judge the coincidence degree of the implied periodicity trend of the data and the conjecture model [18,19,20,21,22]. Consequently, when we use the Lomb–Scargle algorithm, Fourier transform can be applied to non-uniform sampled signals equivalently, which can not only contribute to extract weak periodic signals from the time series, but also to reduce the generation of false signals of the non-uniform time series to some extent. Therefore, the Lomb–Scargle algorithm was selected as the solution method in this experiment. The Lomb–Scargle algorithm is specially used for processing non-uniform sampling signals, and the results acquired also have better unimodality. Moreover, it has been proven that the Lomb–Scargle algorithm can demodulate the non-uniform sampling signal. The algorithm can be described by the following equations:(27)$$\bar{x}=\frac{1}{N}\sum_{j=1}^{N}x_{j},$$
(28)$$\tan(2\omega \tau )=\frac{\sum_{j=1}^{N}\sin(2\omega t_{j})}{\sum_{j=1}^{N}\cos(2\omega t_{j})},$$
(29)$$P(\omega )=\frac{1}{2}\{\frac{[\sum_{j=1}^{N}(x_{i}-\bar{x})\cos\omega (t_{j}-\tau ){]}^{2}}{\sum_{j=1}^{N}\cos^{2}\omega (t_{j}-\tau )}+\frac{[\sum_{j=1}^{N}(x_{i}-\bar{x})\sin\omega (t_{j}-\tau ){]}^{2}}{\sum_{j=1}^{N}\sin^{2}\omega (t_{j}-\tau )}\}$$
where *x_j_* is the *j*th data, *t_j_* is the *j*th time, *τ* is specified for each *ω* to ensure time-shift invariance, and *P*(*ω*) is the power spectral density of the measured data we obtain with the unequal sampling interval for the wavelength. Figure 8 shows the comparison between the frequency spectrum of the interference signal obtained by Fourier transform and the spectrum obtained by the Lomb–Scargle algorithm. Compared with Fourier transform, the result obtained by the Lomb–Scargle algorithm has a better unimodal property. At the same time, compared with the measurement results of the interferometer, this data processing method has high precision. 

As shown in Figure 9, different measuring points were selected in the range of 70 m. The experimental results show that the absolute distance measurement can be realized by dispersive interferometry in the range of 70 m, and the error of measurement was less than 12 μm.

## 4. Conclusions

In this paper, we firstly analyzed different mathematical models of optical frequency comb with an odd or even number of teeth, and derived a unified mathematical model of the optical frequency comb signal, which is not affected by the number of comb teeth. Moreover, we derived the principle of absolute distance measurement based on DPI and proved the feasibility theoretically. At the same time, we compared the frequency spectrum of the interference signal obtained by Fourier transform and the spectrum obtained by the Lomb–Scargle algorithm, and then the latter was used to reduce the error caused by the non-uniform sampling of the spectrometer, which was verified by experiments. 

A long absolute distance measurement experiment with mode-locked OFC was carried out; to overcome the limitations in resolution, the Lomb–Scargle algorithm was adopted to process the data. The distance resolution and the non-ambiguity range of the system are about 10 μm and 0.6 m. The experimental results show that the measurement error was less than 12 μm at the selected measurement points. The results show that the Lomb–Scargle algorithm can effectively reduce the error caused by the resolution, and the error of absolute distance measurement was less than 12 μm in the distance of 70 m based on the mode-locked OFC. Furthermore, owing to the advances in micro-resonator fabrication, there has been a great development in the research of chip-scale soliton microcombs (SMCs) with high repetition frequency, and in the future, we could combine the SMC with the Lomb–Scargle algorithm to eliminate dead zones based on DPI, which will be extended to arbitrary range measurement, as well as applied in ultra-precision semiconductor manufacturing, large aircraft processing, and other precision measurement fields.

## Figures and Tables

**Figure 1 sensors-22-05403-f001:**
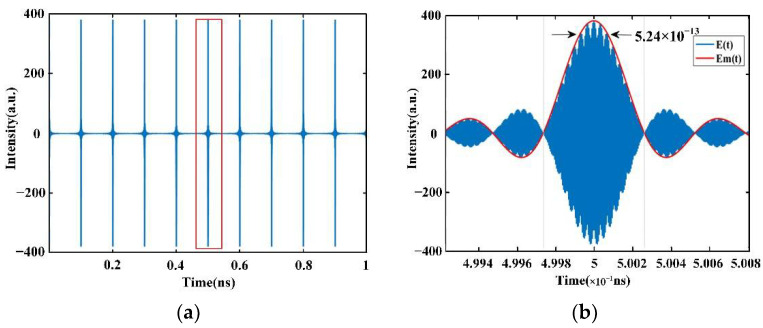
Simulated signal of OFC in the time domain: (**a**) a simplified model of OFC (*M* = 381, *f_r_* = 10 GHz, *f_c_* = 194.2 THz) and (**b**) simulated signal of OFC of one period in the time domain.

**Figure 2 sensors-22-05403-f002:**
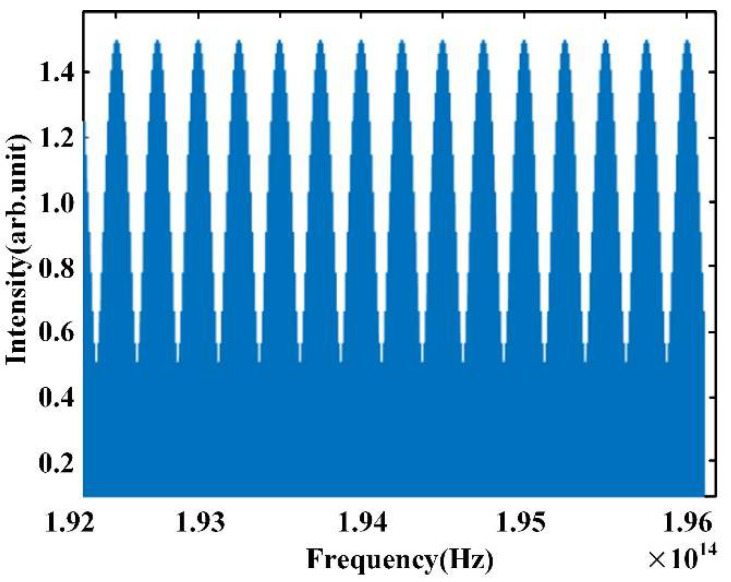
Spectrum of dispersive interferometry.

**Figure 3 sensors-22-05403-f003:**
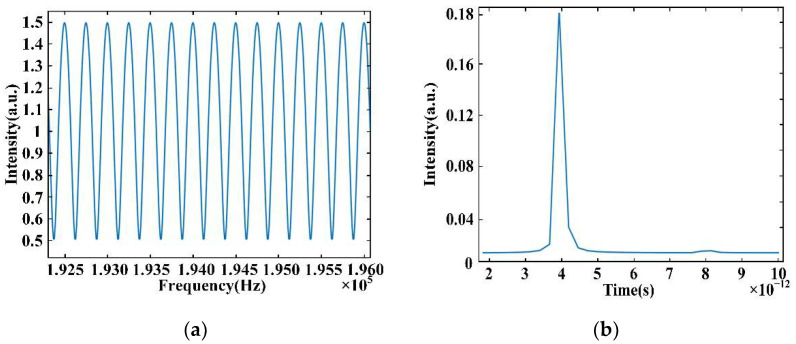
Simulation results of dispersion interference: (**a**) envelope of dispersion interference signal and (**b**) FFT results of dispersion interference.

**Figure 4 sensors-22-05403-f004:**
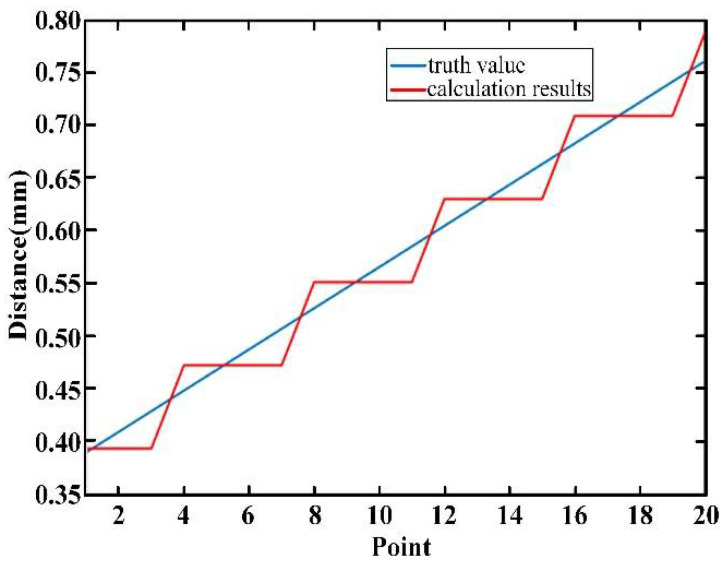
Simulation of resolution of dispersion interference ranging.

**Figure 5 sensors-22-05403-f005:**
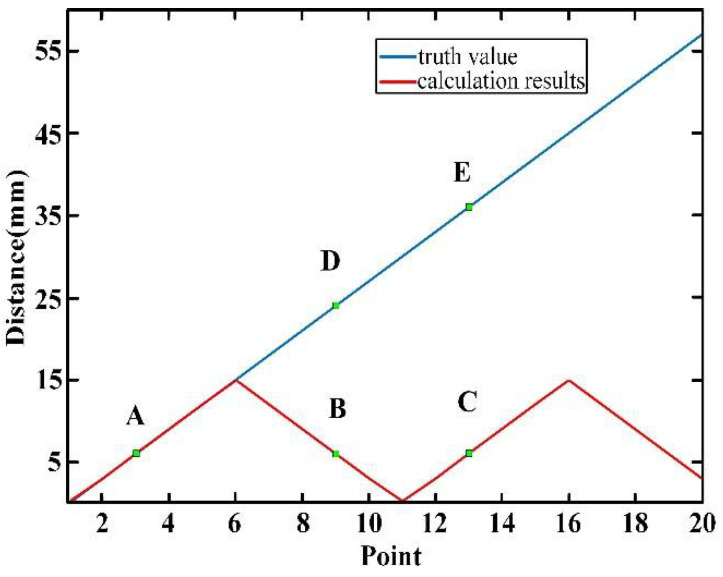
Simulation of non-ambiguity range.

**Figure 6 sensors-22-05403-f006:**
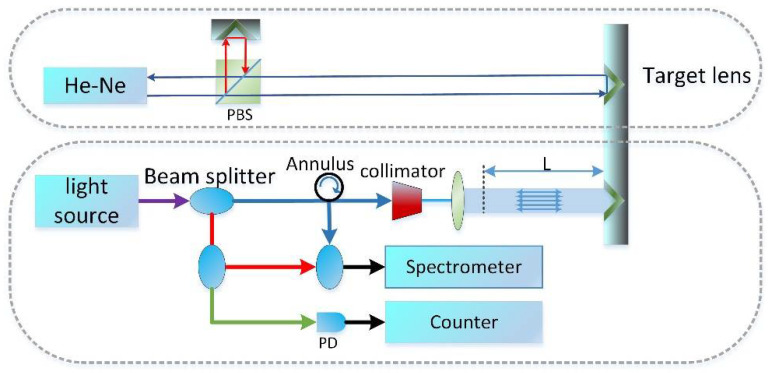
Layout of dispersive interferometry experiment.

**Figure 7 sensors-22-05403-f007:**
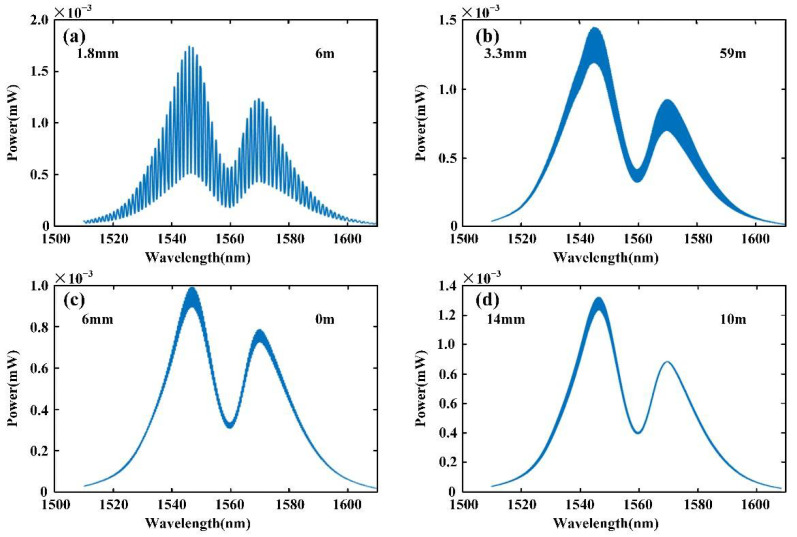
Spectrum of interference signal in experiment: (**a**) the interference spectrum at 1.8 mm; (**b**) the interference spectrum at 3.3 mm; (**c**) the interference spectrum at 6 mm; and (**d**) the interference spectrum at 14 mm.

**Figure 8 sensors-22-05403-f008:**
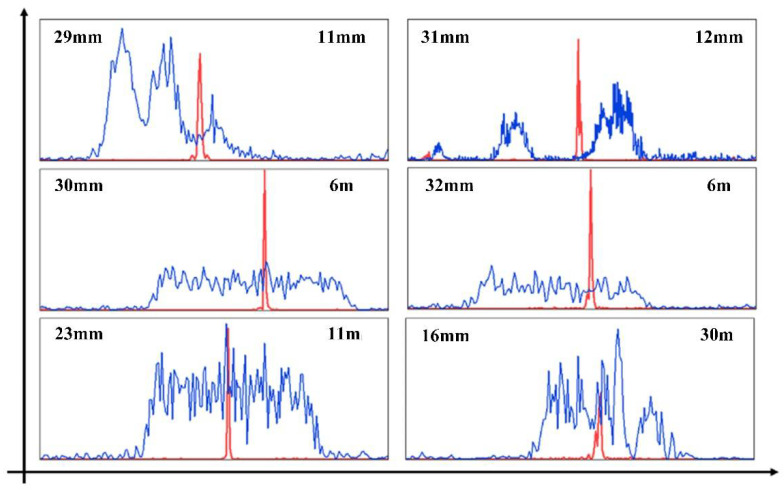
Frequency spectrum of the results obtained by FFT (blue line) and Lomb–Scargle algorithm (red line).

**Figure 9 sensors-22-05403-f009:**
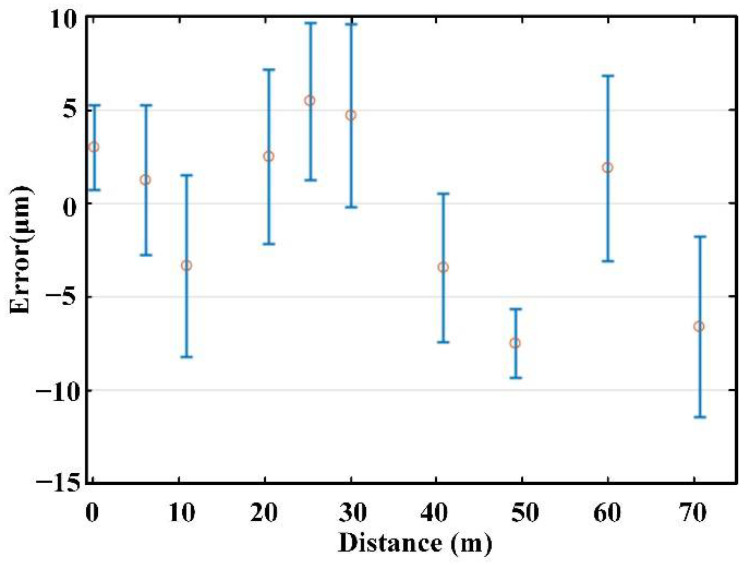
Error diagram of measurement results, where the standard deviations are labeled with error bars (blue lines) and the midpoints of the error bars give the average values of 20 measurements errors at each position (red circles).

## Data Availability

Not applicable.

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
