# Peer review of "Improvement of Distance Measurement Based on Dispersive Interferometry Using Femtosecond Optical Frequency Comb"

_sensors, 2022, doi:10.3390/s22145403_

Round 1

Reviewer 1 Report

Dear Authors,

I think the current submission can be improved in the following ways that I will try to discuss.

1) Thank you for the detailed derivations in Eqs 1-17. However, since it is not the emphasis of the work and covers the well known mathematical model of OFC, I fail to see the significance. All of those discussions could have been given as a part of a supplementary and it could be emphasized in the text that the selection of M being odd or even does not alter the mathematical model of OFC. In my opinion, just a single sentence would be more than enough as this section completely throws the reader off the track and defocuses your emphasis. 

2) Please make sure that the equations are error-free 

a) Eq 16 => rho will be p in the limit

b) Eq 18 => should be squared

c) Eq 20 => will have 2 in front consequently after the square

3) Now, Eqs. 18-25 are very straight forward discussions and equations again. I think your own papers refs 12 and 17 (ref. 12 => please do not cite it as it is shown currently "and so on" by the way in the intro, that is not a good way of authoring a scientific paper). already take a very good care of the Fourier transform approach in depth. Why don't you just refer the reader to those papers in a better way? 

4) I just taught a sophomore class this past semester on signal processing. All of my students are very well aware of the outcomes of Eq. 18-25 to be honest at a sophomore level. They worked on an autocorrelation project to estimate the echo to retrieve the distance measurements. So, these parts should not be presented like the way you give. This paper could have been only a few pages and would not lose much from its content. In the present status, unfortunately the paper discusses very obvious themes at this research level, which degrades the quality of the submission. They could be moved to the supplementary again. 

5) Eq. 30 => deltaf should have been deltalambda. These are very obvious errors. I tried to follow through quickly please make sure the final submission would not have any of these mistakes. 

6)  Now, coming to the main topic of handling Lomb-Scargle Periodograms => this section is the heart of your paper but it is either rushed or very short. It can be lengthened to walk the reader through the details maybe in a similar way that you did earlier for the direct Fourier Analysis. But this is a suggestion only. 

7) Lomb is used as the short form but alternatingly as LOMB or Lomb (stick to one please), why did you need to shorten it? That is very unusual to me.

8) Lomb-Scargle periodograms have been long used in astronomy, it seems. The present submission only has one reference from 70s. I would strongly recommend adding these to the references:

a) https://doi.org/10.3847/1538-4365/aab766

b) https://doi.org/10.5194/amt-13-467-2020

c) https://doi.org/10.1016/j.epsr.2021.107251 (outside of astronomy)

d) http://dx.doi.org/10.1063/1.1381888

e) Thesis work by Connor Frederick (Enabling Precision Astronomical Spectroscopy with Laser Frequency Combs - ProQuest) => I think you have submitted just in time because of his possible overlap with OFCs and Lomb analysis. 

9) It really is timely to adapt Lomb-Scargle periodograms to OFCs as other fields are following the same trend as in 8)c). However, please revise your abstract and conclusion as it sounds like the authors have found this method. In contrast, it is an adaptation of a well known method. The reader should get this clearly. It is very clear that the straight forward Fourier transform method will be limited by Nyquist limit. Every engineer should know this in my opinion. It could be rewritten like "Hence, to overcome the limitations in resolution we have adapted Lomb-Scargle periodograms relying on least squares method".

10) Please make sure that a native reader takes a look at the submission as there are a few typos or grammer mistakes => eg. were got instead of obtained or estimated etc. 

Best Regards,

Author Response

Dear  reviewers of Sensors:

     Thanks to the editors and reviewers in a busy schedule to review the manuscript. All your suggestions are very important, which have vital guiding significance for my thesis writing and scientific research work. Based on the opinions of the reviewers, we have reply to them one by one and make revisions in our manuscript.

Reviewer 2 Report

Dear authors

This is an interesting research using both a mathematic model and experiment to demonstrate the enhancement of the distance measurement using optical frequency comb (OFC). The concept of manuscipt is novel, but its design and structure seem to be not appropriate for a research paper. Thus, it needs to be upgraded and revised following major problems before publication.

1. All contents of the manuscript should be reorganized in the normal form of a reseach paper to aid the readers. It should present basic sections including: Introduction, Methods and Experiments, Results and discusions, and conclusions

2. Because this manuscript used a mathematic model, it has many mathematic equations. However, the main content of manuscript should show the important results, the others should be put in the supporting information.

3. The authors mainly showed the results and there is no discussion and comparison, which didnot emphasize the novelty and importance of this reseach

4. In the introduction, the previous reseaches in the literature are only listed. it should be comparized and indicated the limitations and problems

Author Response

(The authors gave the same response as above.)

Reviewer 3 Report

This paper proposed new mathematical model with OFC and DPI, and applied Lomb-Scargle algorithm (LOMB) which can reduce the error of un-uniform sampling data. The error of absolute distance measurement is reduced with LOMB with long-range distance. I think this paper is acceptable for “Sensors”. Some minor comments are listed below.

1.    The authors demonstrated LOMB for absolute distance measurement for long-range (70 m). Then, does LOMB also effective to small-range condition?

2.    For figure 9, the unit of distance is mm. Please check the unit of distance.

Author Response

(The authors gave the same response as above.)

Round 2

Reviewer 2 Report

I have checked the manuscript. The authors reflected all of my comments. Thus I think it can be accepted for publication